# Identification of key sectors of water resource utilization in China from the perspective of water footprint

**Guangyao Deng[1], Xiaofang Yue[2]\*, Lu Miao[2], Fengying Lu[1]**

**1** School of Statistics, Lanzhou University of Finance and Economics, Lanzhou, PR China, **2** China Center for Special Economic Zone Research, Shenzhen University, Shenzhen, PR China

\* harborangel@163.com

**Data Availability Statement:** Other researchers may freely access the EORA National IO Tables (Chinese IO Tables) from The Eora Global Supply Chain Database at https://www.worldmrio.com/. Users must first register for an account to access

## Abstract

We identified the key sectors of water resource use in China from the perspective of the water footprint to improve the use of water resources. The empirical results showed that there were six key sectors (including Crop Cultivation; Forestry; Livestock and Livestock Products; Fishery, Technical Services for Agriculture, Forestry, Livestock and Fishing; Other Food Products, and Scrap and Waste) for water consumption in China in 2015.We analyzed the use of green water, blue water, and grey water. These six sectors accounted for 66.15% of the total impact and 90.76% of the direct impact. Seven key sectors (the six sectors above plus Steel Processing)for the consumption of blue water in China can explain 59.70% of the total impact and 86.94% of the direct effect in 2015. Eight key sectors (Crop cultivation, Other food products, Scrap and Waste, Railway Freight Transport, Highway Freight and Passengers Transport, Water Freight and Passengers Transport, Pipeline Transport, and Health Services) responsible for the consumption of grey water in China in 2015 can explain 81.28% of the total impact and 95.73% of the direct impact. Therefore, the Chinese government should focus on the departments that manage water resources in these sectors when designing water-saving policies and improving water-use efficiency, such as promoting water-saving irrigation technology (including sprinkler irrigation and drip irrigation) in the agricultural sector.

## Introduction

Water, the source of life, drives productivity. China's demand for production water and domestic water has increased significantly due to its rapid economic development and population growth [1–5]. However, the spatial–temporal distribution of water resources is unbalanced. Therefore, there is a serious imbalance between the supply and demand for water resources. Since there are multiple sectors in any economic system, it is critical to identify which sectors deserve special attention to better manage the limited water resources. In this study, we argue that the key sectors of water resource use are not only the sectors with large water resource consumption but also the sectors where water resource use can greatly promote

the tables. Additional questions or access requests may be sent to info@worldmrio.com.

**Funding:** This work was supported by the Natural Science Foundation of China under Grant [number 71704070]; Ministry of Education for the Humanities and Social Sciences Research Young Fund on the West and Borderland Project [number 17XJC790002]; Guangdong Provincial Natural Science Foundation of China under Grant [number 2017A030313443];and Program of Lanzhou University of Finance and Economics under Grant [number Lzufe2018B-06].

**Competing interests:** The authors have declared that no competing interests exist.

economic growth. Within the production process of each sector, the water consumption includes not only the direct approach but also the indirect approach, that is, water is consumed to produce intermediate productions that are the raw materials for other sectors. Therefore, the key sectors of water resource use need to be identified from the perspective of the water footprint [6].

The identification of key sectors can be traced back to Rasmusen [7] and Hirschman [8]. Based on the input–output model, they defined the sensitivity coefficient and the influence coefficient to describe the role of each sector in the economic system. When the sensitivity coefficient and the influence coefficient of a certain sector are greater than 1, the sector is considered to play a greater role in the economic system and is deemed a key sector. Using various studies from the literature, Alcántaraand Padilla [9] established an input–output model to identify the key sectors of the final energy consumption. They identified the key sectors of Spanish energy consumption based on the demand elasticity of the final energy consumption. Alcántaraand Padilla [10] analyzed the identification of key sectors for carbon emissions from the producers' perspective and the value-added angle using Spain as an example for their empirical analysis. Othman and Jafari [11] applied the model proposed by Alcántara and Padilla in 2006 [10] and identified the key sectors of carbon emissions in terms of production in Malaysia in 2005. We drew lessons from the researchers noted above to study the identification of the key sectors of water use in China from the perspective of the water footprint to realize the maximum economic value of the water resources and allocate these water resources in a reasonable way in light of the nation's limited water supply.

The water footprint refers to the sum of the direct and indirect water use by a country (region or individual) for production (consumption) [6] [12] [13]. It can be divided into a green water footprint, blue water footprint, and grey water footprint. Green water refers to the water stored in the soil at the root of crops (mainly rain water), blue water refers to rivers, lakes, and groundwater with economic value, and grey water refers to the water needed to purify pollutants in the water. Virtual water refers to the water required in the production and service process [14–16]. Scholars have applied the input–output model to study issues in water resources, such as the water footprint and virtual water. Some researchers studied the water footprint and virtual water trade in China (or other countries) or a specific province with the single-region input–output model [17–24]. Many researchers studied the water footprint and virtual water trade in multiple regions or provinces of China (or other countries) with the multi-region input–output model [25–34].

The focus of this study is as follows: Inspired by the model of identifying key sectors of carbon emissions developed by Alcántaraand Padilla [8], we explore the key sectors of green, blue, and grey water consumption in China from the perspective of the water footprint and discuss the corresponding policy recommendations. The key sectors of water use defined in this paper are not only the sectors with a large amount of water use, but also the sectors where water resources can drive economic growth to a large extent. Therefore, we used the total impact (TI) indicator and direct impact (DI) indicator as analysis tools. We consider the percentage change in the water use in various sectors caused by a 1% increase in the value added of sector, *j* (TI),as well as the percentage change in the water use in sector i when China's GDP (the total value added by all sectors) increased by 1% [9], [10].

The existing literature on the water footprint focuses on its accounting, which can only identify the key sectors that use water and the amount, but cannot identify the sectors where water resource input can greatly drive economic growth. The innovation of this paper is that it identifies the key sectors of water resource use in terms of the amount of water used and the economic function of that water use by drawing on the method proposed by Alcántara and Padilla [10], which is used to identify the key sectors of carbon emissions. It should be pointed

out that the purpose of this study was to identify the key sectors of China's water use in combination with water use and economic functions, not to account for the green, blue, and grey water footprints of various sectors in China. The data on green water, blue water, and grey water footprints of various sectors in China are from the open EORA database (https://www.worldmrio.com/). The specific calculation method of the data in this database can be found in the literature (Lenzen et al. [35] and Lenzen et al. [36]).

## Method

We first defined the direct consumption coefficient matrix, as shown below:

$$
\mathbf{A} = \begin{bmatrix} a_{11} & a_{12} & \cdots & a_{1n} \\ a_{21} & a_{22} & \cdots & a_{2n} \\ \cdots & \cdots & \cdots & \cdots \\ a_{n1} & a_{n2} & \cdots & a_{nn} \end{bmatrix},
\tag{1}
$$

where n is the total number of sectors in the input–output table, $a_{ij} = X_{ij}/X_j$, $X_{ij}$ is the amount of products of sector i (intermediate input) that need to be consumed in order to produce the products of sector j, and $X_j$ is the total input of sector j.

By closely following the notations and frameworks constructed by Alcántaraand Padilla [9], Alcántaraand Padilla [10], and Othman and Jafari [11], we found that (without regard to imports and exports) there exists the following relationship between the total investment, the intermediate input, and value added of each sector:

$$
\mathbf{X} = \widehat{\mathbf{X}} \mathbf{A}' \mathbf{u} + \mathbf{v},
\tag{2}
$$

where $\mathbf{X}$ is the total input row vector of the order $n \times 1$, $\widehat{\mathbf{X}}$ is the diagonalized matrix of $\mathbf{X}$, $\mathbf{A}$' is the transposed matrix of the direct consumption coefficient matrix, $\mathbf{u}$ is the unit row vector of order $n \times 1$, and $\mathbf{v}$ the value added row vector of the order $n \times 1$. We multiplied both sides of Eq (2) by $\widehat{\mathbf{X}}-1$ and obtained:

$$
\mathbf{u} = \mathbf{A}' \mathbf{u} + \mathbf{s}, \mathbf{s} = \widehat{\mathbf{X}}^{-1} \mathbf{v},
\tag{3}
$$

where $\mathbf{s}$ is the $n \times 1$ value-added coefficient row vector. Eq (3) is converted into

$$
\mathbf{u} = (\mathbf{I} - \mathbf{A}')^{-1} \mathbf{s}
\tag{4}
$$

We defined $g_j$, the water consumption share of sector j as follows:

$$
\mathbf{g}_j = \frac{w_j}{W},
\tag{5}
$$

where $w_j$ represents the water consumption of sector j (the consumption of green, blue, and grey water), and W is the total water consumption of all sectors. By combining Eqs (4) and (5), we obtained:

$$
\mathbf{w} = \widehat{\mathbf{w}} (\mathbf{I} - \mathbf{A}')^{-1} \mathbf{s}
\tag{6}
$$

$$
\mathbf{w} = W \widehat{\mathbf{g}} (\mathbf{I} - \mathbf{A}')^{-1} \mathbf{s},
\tag{7}
$$

where $\mathbf{w}$ is the $n \times 1$ row vector composed of the water consumption of all sectors, and $\widehat{\mathbf{w}}$ is the

diagonal matrix obtained by the diagonalizing vector **w**. We also diagonalized the share coefficient vector **g**' = $(g_1, g_2, \cdots, g_n)$ to get the diagonal vector $\widehat{\mathbf{g}}$ We multiplied both sides of Eq (7) by **u**' to obtain:

$$W = W\mathbf{g}'(\mathbf{I} - \mathbf{A}')^{-1}\mathbf{s}. \tag{8}$$

By writing Eq (8) in an incremental form, we obtained:

$$\Delta W = W\mathbf{g}'(\mathbf{I} - \mathbf{A}')^{-1}\mathbf{s}\alpha, \tag{9}$$

where $\alpha$ is the proportion of the increase of the value added. We divided both sides of Eq (9) by $W$ and obtained:

$$W^{-1}\Delta W = \mathbf{g}'(\mathbf{I} - \mathbf{A}')^{-1}\mathbf{s}\alpha, \tag{10}$$

Furthermore, by diagonalizing **s** in Eq (9), we obtained

$$\boldsymbol{\varepsilon}' = \mathbf{g}'(\mathbf{I} - \mathbf{A}')^{-1}\widehat{\mathbf{s}}\,\alpha, \tag{11}$$

where **ε**' actually represents the elasticity between the amount of water used and the value added, that is,5

$$\varepsilon_i = \frac{\Delta W}{W}\frac{v_i}{\Delta v_i}. \tag{12}$$

By diagonalizing **g**' in Eq (11), and letting $\alpha = 1\%$, we obtained:

$$\mathbf{W^v} = \widehat{\mathbf{g}}(\mathbf{I} - \mathbf{A}')^{-1}\widehat{\mathbf{s}}, \tag{13}$$

where the economic meaning of $W_{ij}^v$, which is the representative element of the matrix $\mathbf{W^v}$, is the increase of the water resources used in sector $i$ caused by the 1% increase of the value added of sector $j$, namely, the cross-elasticity between the water use and the value added. We further defined the TI (total impact) indicator (Eq 14) and DI (direct impact) indicator (Eq 15) using the literature [9], [10]:

$$\mathrm{TI}_j = \sum_{i=1}^{n} W_{ij}^v \tag{14}$$

$$\mathrm{DI}_i = \sum_{j=1}^{n} W_{ij}^v. \tag{15}$$

The economic implication of $TI_j$ is the percentage change in the water use in various sectors caused by a 1% increase in the value added to sector $j$. The economic implication of $DI_i$ is the percentage change in the water use in sector i when China's GDP (the total value added by all of the sectors) increases by 1%. In addition, according to Alcántara and Padilla [10] and Othman and Jafari [11], both the sum of the total impact of each sector and the sum of the direct impact of each sector are equal to 1, that is, $\sum_{j=1}^{n} TI_j = 1$ and $\sum_{i=1}^{n} DI_i = 1$.

Due to the large number of sectors involved (99), we employed 0.003 as the threshold to classify all sectors based on the calculated results. Specifically, if $TI \geq 0.003$ and $DI \geq 0.003$, it was regarded as a key sector of water consumption; if $TI < 0.003$ and $DI \geq 0.003$, it was regarded as a direct impact dominant sector; if $TI \geq 0.003$ and $DI < 0.003$, it was treated as a total impact

dominant sector, and if *TI*<0.003 and *DI*<0.003, we treated it as a non-key sector of water consumption. It should be noted that Alcántara and Padilla [9], Alcántara and Padilla [10], Othman and Jafari [11] and other researchers have employed the median as the classification standard. Considering the characteristic that China's water use is mainly concentrated in the sector Crop Cultivation (Sector1), we employed 0.003 as the threshold for classification, which ensured that the total impact (TI) and direct impact (DI) of the key sectors account for more than 50% of that of various sectors. In addition, since the input–output table used in this study has 99 sectors, the number of key sectors identified by the median as a threshold would be too great.

## Data

We obtained the data from the national input–output table in the Eora database [35], [36]. The Eora database contains the input–output tables for each year in China from 1970 to 2015, calculated using the producer price and the buyer price, respectively. We selected the producer price-based input–output table for 123 sectors in 2011–2015 as the research object.

The input–output table used in the research methodology section of this paper does not include the imports and exports (i.e., the table does not involve the import and export trade;it is known as an import-included competitive input–output table), while the national input–output table provided in the Eora database was an import-included non-competitive input–output table (i.e., imports and exports were included). Therefore, to be consistent with the research methodology, in this paper, we define the total input as the sum of the intermediate input and the value added and the total output is equal to the intermediate use plus the final use. The final use here includes the original final use of the input–output table of China in the Eora database, the net export (which is the exports minus the imports), and the error term (the correction term is introduced in order to ensure that the sum of the input of all the sectors is equal to the total output.).

The green water footprint, blue water footprint, and grey water footprint of each sector were derived from China's input–output table in the Eora database. Since the raw data of the green water footprint, blue water footprint, and grey footprint of some sectors, such as the Technical Services for Agriculture, Forestry, and Livestock, and Fishing, were all 0, which caused the matrix $\hat{\mathbf{g}}$ to be not invertible, we merged these sectors and their previous one with non-zero water footprints to form a combined sector. The total number of sectors after merging was 99 (see the S1 Appendix for the specific name of each sector). In addition, Water Production and Supply, Construction, and all industries in the tertiary industry had a green water footprint of 0. Thus, we only considered identification of the use of blue water footprint, grey water footprint, and sum of the 3 water footprints (the green water footprint, blue water footprint, and grey water footprint) of the key sectors. In order to keep the total number of studied sectors consistent, we did not evaluate the key sectors of green water use alone, as the amount of green water used for most sectors in China in 2011–2015 was 0, especially for the sectors in the tertiary industries that consume no green water (if we wanted to evaluate the key sectors of green water use alone, we would merge more sectors).

## Results and discussion

We first organized the water footprint data to analyze the industrial characteristics and trends to introduce the basics of water consumption in China so that readers can understand its water footprint; we then conducted the studies on the key sectors of China's water use.

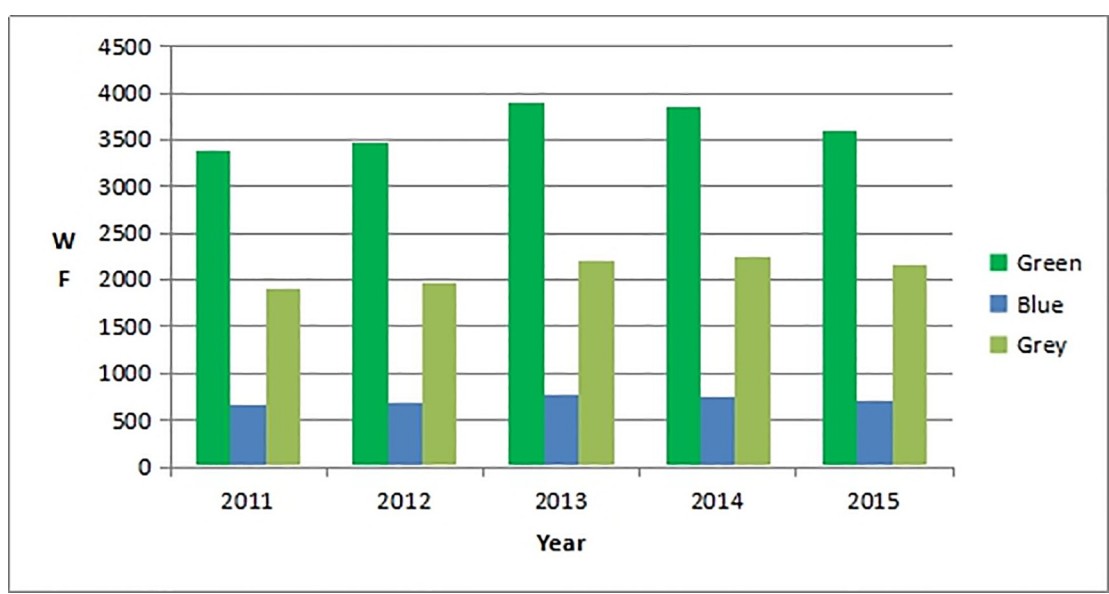

**Fig 1. Green water footprint, blue water footprint, and grey water footprint of China in 2011–2015.** (: WF is the water footprint; the units are in billions of cubic meters.).

### Trend analysis and industrial characteristics of the water footprints

The green water footprint, blue water footprint, and grey water footprint of China in 2011–2015 are shown in Fig 1 (the water footprint in this study only includes the production water; the domestic water was excluded).

Fig 1 shows that China's green water footprint was the largest among the footprints, followed by its grey and blue water footprints in 2011–2015. The green water footprint and blue water footprint increased in 2011–2013, but declined in 2013–2015. The grey water footprint increased in 2011–2014, but declined in 2014–2015.

Table 1 shows the data for 2015 and illustrates the use of green water, blue water, and grey water in three major industries.

Table 1 shows that compared with the secondary industry and the tertiary industry, the primary industry has a greater water footprint. In 2015, China's green water footprint was the greatest, followed by its grey and blue water footprints. These three types of water footprints in the tertiary industry were small, and the green water footprint was 0 (the blue water footprint and the grey water footprint of the tertiary industry in 2015 were 7 billion cubic meters and 86 billion cubic meters, respectively).

We were not able to represent each data point individually in the graph because of the massive amount of data (please see the attachments for specific data). This is why we only listed the values of the green water, blue water, and grey water footprints in all sectors of China in 2011–2015 (Fig 1), as well as the water footprint value in three major industries in 2015 (Table 1).

**Table 1. Green water footprint, blue water footprint, and grey water footprint inthree major industries in China in 2015 (unit: billions of cubic meters).**

| Water Footprint | Primary | Secondary | Tertiary |
|---|---|---|---|
| Green | 3063 | 531 | 0 |
| Blue | 576 | 121 | 7 |
| Grey | 1057 | 1024 | 86 |

## The key consumption sectors of the green water, blue water, and grey water

We used the classification criteria TI≥0.003 and DI≥0.003. The key sectors of the water resource consumption of China in 2015 are shown in Table 2 (the sum of the green water, blue water, and grey water).

Table 2 shows that six sectors, which include Crop Cultivation (S1), Forestry (S2), Livestock and Livestock products (S4), Fishery, Technical Services for Agriculture, Forestry, Livestock, and Fishing (S5), Other Food Products (S18), and Scrap and Waste (S83), account for 66.15% of the TI, and also explain 90.76% of the DI. The remaining 93 sectors can only explain 33.85% of the TI and 9.24% of the DI, so the above six sectors can be called the key sectors of China's water consumption in 2015 (green water, blue water, and grey water). Since the production of crops such as rice, wheat, corn, beans, and potatoes requires a great amount of water resources, the increase in the value added of the Crop Cultivation sector by 1% would lead to the increase in the sum of the water use in various sectors by 0.6615% [10]. In addition, the correlation effect exists between the Crop Cultivation sector and other sectors [2], that is, the production of other sectors requires the products of Crop Cultivation as an intermediate input, so a 1% increase of the sum of the value added of all the sectors would lead to a 0.7008% increase in the water consumption of the Crop Cultivation sector. The production of forest products, animal products, fisheries, and food processing products also requires a great amount of water resources. Therefore, the TI and DI of Forestry (S2), Livestock and Livestock Products (S4), Fishery, Technical Services for Agriculture, Forestry, Livestock, and Fishing (S5), and Other Food Products (S18) are also large. In order to purify the pollutants from the water, the Scrap and Waste (S83) sector uses a large amount of grey water, so the TI and DI of the Scrap and Waste sector are also large.

We examined whether the TI and DI are greater than 0.003 as the threshold to study the key sectors of China's water resources (green water, blue water, and grey water) use in 2011–2015, and observed the changes in the TI and DI of these key sectors. The specific results are shown in Table 3.

Table 3 shows that with a threshold of 0.003 in 2011–2013, S2 (Forestry) is not a key sector anymore, indicating that the key sectors may change over time. For S1 (Crop Cultivation), S5 (Technical Services for Agriculture, Forestry, Livestock, and Fishing), and S83 (Scrap and Waste), the TI surpasses the DI in 2011–2015; for S18 (Other Food Products) in 2011–2015, the value of the TI is smaller than that of the DI. For S4 (Livestock and Livestock Products), the value of the TI is greater than the value of the DI in 2011–2014, but in 2015, the total

**Table 2. Key sectors of the water resource consumption (green water, blue water, and grey water) of China in 2015.**

| Sector | TI | DI |
|--------|------|------|
| S1 | 0.4681 | 0.7008 |
| S2 | 0.0041 | 0.0034 |
| S4 | 0.0157 | 0.0164 |
| S5 | 0.0127 | 0.0050 |
| S18 | 0.0104 | 0.0335 |
| S83 | 0.1505 | 0.1485 |
| Total | 0.6615 | 0.9076 |

Food production includes the following sectors: Grain Mill Products (S12),FoodStuff Production and Processing (S13), Vegetable Oils and Forage (S14), Sugar Refining (S15), Slaughtering, Meat Processing, Eggs and Dairy Products (S16), Prepared Fish and Seafood (S17),and Other Food Products (S18).

**Table 3. The Total Impact (TI) and Direct Impact (DI) of the use of water resources (green water, blue water, and grey water) of key sectors in China in 2011–2015.**

| Sector | 2011 | | 2012 | | 2013 | | 2014 | | 2015 | |
|---|---|---|---|---|---|---|---|---|---|---|
| | TI | DI | TI | DI | TI | DI | TI | DI | TI | DI |
| S1 | 0.5394 | 0.7296 | 0.5419 | 0.7278 | 0.5471 | 0.7290 | 0.5053 | 0.7182 | 0.4681 | 0.7008 |
| S2 | 0.0037 | 0.0029 | 0.0036 | 0.0029 | 0.0036 | 0.0029 | 0.0038 | 0.0031 | 0.0041 | 0.0034 |
| S4 | 0.0163 | 0.0156 | 0.0162 | 0.0157 | 0.0163 | 0.0157 | 0.0161 | 0.0160 | 0.0157 | 0.0164 |
| S5 | 0.0105 | 0.0048 | 0.0104 | 0.0048 | 0.0103 | 0.0048 | 0.0114 | 0.0049 | 0.0127 | 0.0050 |
| S18 | 0.0075 | 0.0310 | 0.0078 | 0.0312 | 0.0078 | 0.0310 | 0.0092 | 0.0319 | 0.0104 | 0.0335 |
| S83 | 0.1306 | 0.1288 | 0.1335 | 0.1316 | 0.1330 | 0.1312 | 0.1409 | 0.1390 | 0.1505 | 0.1485 |
| Total 1 | 0.7043 | 0.9099 | 0.7098 | 0.9110 | 0.7145 | 0.9116 | 0.6829 | 0.9099 | 0.6573 | 0.9042 |
| Total 2 | 0.7080 | 0.9127 | 0.7134 | 0.9139 | 0.7181 | 0.9145 | 0.6867 | 0.9130 | 0.6614 | 0.9076 |

Total 1 does not include the value of Sector 2; Total 2 includes the value of Sector 2.

impact is smaller than the direct impact, that is, TI<DI. In total, the value of the TI is smaller than the value of the DI each year, which indicates that the total impact was more dispersed than the direct impact.

The key sectors of China's water use varied from year to year. Table 2 shows that for sector S2 (Forestry), the DI indicator in 2011–2013 is less than 0.003, which does not conform to TI>0.003 and DI>0.003, so it cannot be identified as a key sector. However, in 2014–2015, sector S2 (Forestry) meets the conditions of TI>0.003 and DI>0.003, and thus, it was identified as a key sector. In the EORA database, the water usage of each sector in China varied from year to year, and there were different input–output tables each year. Therefore, even for the same sector, the DI indicator and TI indicator varied from year to year. The key sectors of China's water use can change over time.

We used 2015 as an example. We analyzed the sectors where the direct impact dominates (TI<0.003 and DI≥0.003) and the sectors where the total impact dominates (TI≥0.003 and DI<0.003), and the specific results are illustrated in Table 4.

Table 4 indicates that the direct impact dominant sectors include Vegetable Oil and Forage (S14), Slaughtering, Meat Processing, Eggs, and Dairy Products (S16), Prepared Fish and Seafood (S17), and Arts and Crafts Products (S81). Except for S81, the other 3 sectors belong to the food processing industry. The total impact dominant sectors are Coal Mining and Processing (S6), Crude Petroleum Products and Natural Gas Products (S7), and Non-metal Minerals and Other Mining (S11), which belong to the mining industry. Sectors in the manufacturing industry, including Grain Mill Products(S12), Petroleum Refining(S35), Raw Chemical Materials(S37), Chemical Fertilizers(S38), Chemical Pesticides(S39), Plastic Products(S47), Steel

**Table 4. Direct Impact (DI) dominant sectors and Total Impact (TI) dominant sectors in 2015.**

| Sector | TI | DI | Sector | TI | DI | Sector | TI | DI |
|---|---|---|---|---|---|---|---|---|
| S6 | 0.0101 | 0.0006 | S37 | 0.0045 | 0.0010 | S81 | 0.0019 | 0.0039 |
| S7 | 0.0157 | 0.0004 | S38 | 0.0297 | 0.0007 | S84 | 0.0233 | 0.0021 |
| S11 | 0.0037 | 0.0003 | S39 | 0.0052 | 0.0002 | S89 | 0.0067 | 0.0013 |
| S12 | 0.0030 | 0.0008 | S47 | 0.0160 | 0.0009 | S90 | 0.0132 | 0.0013 |
| S14 | 0.0028 | 0.0058 | S55 | 0.0043 | 0.0026 | S92 | 0.0090 | 0.0013 |
| S16 | 0.0018 | 0.0068 | S59 | 0.0037 | 0.0017 | S95 | 0.0338 | 0.0014 |
| S17 | 0.0015 | 0.0036 | S62 | 0.0049 | 0.0017 | S96 | 0.0439 | 0.0010 |
| S35 | 0.0067 | 0.0020 | S67 | 0.0031 | 0.0012 | S97 | 0.0070 | 0.0009 |

Processing(S55), Metal Products(S59), Other General Industrial Machinery (S62), and Vehicle Fittings Production (S67), are also total impact dominant sectors. In addition, the total impact plays a dominant role in the Electricity and Steam Production and Supply (S84)sector as well as some service industry sectors, such as Railway Freight Transport (S89), Highway Freight and Passenger Transport (S90), Water Freight and Passenger Transport (S92), Pipeline Transport (S95), Hotels (S96), and Water Conservancy (S97). S95–S97 are merged sectors. According to the descriptions from the data source, the consumption of blue water and grey water only existed in sectors such as Pipeline Transport, Hotels, and Water Conservancy before they were merged. The total impact dominant sectors inS95–S97 are Pipeline Transport, Hotels, and Water Conservancy.

Using the statistics in Table 4, we obtained the sum of the TI values of the above 24 sectors (0.2555) and the sum of the DI values (0.0435). Combined with the data of the 6 key sectors in Table 1, the sum of the TI values of these 30 sectors is 0.9170, and the sum of the DI values is 0.9511, that is, the above 30 sectors explain 91.70% of the total impact, while explaining 95.11% of the direct impact. Therefore, the remaining 69 sectors can be called non-key sectors of water use.

## The key sectors of consumption of blue water or grey water

Although Tables 2 to 4 reveal the sum of the consumption of green water, blue water, and grey water in various sectors in China, we did not analyze the consumption of blue water or grey water separately. We analyzed the key sectors of blue or grey water use. We followed the criteria given in Section 4.1 and still employed the threshold of 0.003. When TI>0.003 and DI>0.003, the corresponding sector was called a key sector of blue water or grey water use. The empirical results for 2011–2015 are shown in Tables 5 and 6.

We can see from Table 5 that the most important key sector for the use of blue water in China in 2011–2015 is Crop Cultivation (S1), which accounts for 52.87%–60.81% of the TI, and 79.21%–81.08% of the DI. Moreover, both values of the TI and DI of the Crop Cultivation (S1) sector in Table 5 are higher than those in Table 3. Compared with the data in Table 3, the values of the TI and DI of the Scrap and Waste (S83) sector in Table 5 dropped sharply, which is due to the fact that the Sector Scrap and Waste sector mainly uses grey water and its consumption of blue water is relatively small. There is one more key sector that is important; the Steel Processing (S55)sector uses a significant amount of blue water but little green water and grey water (compare the values shown in Table 5 and Table 3). In terms of totals, the TI is smaller than the DI each year. In Table 5, in the period of 2011–2015, the TI of the 7 key sectors accounted for 59.70%–66.81% of the TI of 99 sectors, and the DI of those 7 sectors accounted for 86.94–88.11%, so they can be called key sectors of blue water consumption.

Table 6 shows that the main sectors for the use of grey water in China in 2011–2015 are Crop Cultivation (S1) and Scrap and Waste (S83). Since the purification of the residues of chemical products such as pesticides and fertilizers consumes a large amount of water resources, the Crop Cultivation (S1) and Scrap and Waste (S83) sectors use more grey water, and thus they have relatively high TI and DI values. In addition to Crop Cultivation (S1) and Scrap and Waste (S83), the key sectors for grey water use in 2011–2014 are Other Food Products (S18), Railway Freight Transport (S89), Highway Freight and Passengers Transport (S90), Water Freight and Passengers Transport (S92), Pipeline Transport (S95), and Health Services (S99). In 2015, Railway Passenger Transport (S88) was also included. S95 and S99 are merged sectors. We selected the sectors that had original grey water values above 0. From the aggregate perspective, the above sectors account for more than 80% of the TI, as well as more than 95% of the DI of 99 sectors. Therefore, they can be called key sectors of grey water use.

Table 5. The key sectors of the consumption of blue water in China in 2011–2015.

| Sector | 2011 | | 2012 | | 2013 | | 2014 | | 2015 | |
|---|---|---|---|---|---|---|---|---|---|---|
| | TI | DI | TI | DI | TI | DI | TI | DI | TI | DI |
| S1 | 0.5975 | 0.8088 | 0.6023 | 0.8095 | 0.6081 | 0.8108 | 0.5660 | 0.8051 | 0.5287 | 0.7921 |
| S2 | 0.0046 | 0.0036 | 0.0046 | 0.0036 | 0.0045 | 0.0036 | 0.0048 | 0.0039 | 0.0052 | 0.0043 |
| S4 | 0.0166 | 0.0151 | 0.0166 | 0.0152 | 0.0166 | 0.0152 | 0.0165 | 0.0157 | 0.0162 | 0.0162 |
| S5 | 0.0107 | 0.0040 | 0.0106 | 0.0040 | 0.0104 | 0.0040 | 0.0118 | 0.0041 | 0.0132 | 0.0043 |
| S18 | 0.0080 | 0.0326 | 0.0083 | 0.0329 | 0.0083 | 0.0327 | 0.0098 | 0.0339 | 0.0112 | 0.0359 |
| S55 | 0.0054 | 0.0034 | 0.0053 | 0.0034 | 0.0052 | 0.0034 | 0.0054 | 0.0033 | 0.0055 | 0.0034 |
| S83 | 0.0149 | 0.0112 | 0.0152 | 0.0115 | 0.0150 | 0.0114 | 0.0160 | 0.0122 | 0.0170 | 0.0132 |
| Total | 0.6577 | 0.8787 | 0.6629 | 0.8801 | 0.6681 | 0.8811 | 0.6303 | 0.8782 | 0.5970 | 0.8694 |

The key sectors of the water consumption can change for the different types of water footprints. For instance, according to Table 5 and Table 6, the key sectors that used blue water are different from those with grey water. As the functions of blue water and grey water are not the same, the first is for the production and services of each sector while the second is for removing pollutants from water, sectors that use more blue water do not necessarily consume more grey water. In addition, how the blue water use of other sectors changes (which is caused by the change in the value added of each sector) is different from the change in the grey water use.

We further analyzed the total impact dominant sector and direct impact dominant sector for the blue water footprint and grey water footprint by employing the data from 2015, and the results are shown in Table 7 and Table 8.

Table 7 shows that according to the standards TI<0.003 and DI>0.003, the direct impact dominant sectors of blue water consumption in China in 2015 are as follows: Vegetable Oil and Forage (S14), Slaughtering, Meat Processing, Eggs, and Dairy Products (S16), Prepared Fish and Seafood (S17), and Arts and Crafts Products (S81). They all belong to the food processing industry except for S81. In accordance with the standard of TI≥0.003 and DI<0.003, the total impact dominant sectors of blue water consumption in China in 2015 include Coal Mining and Processing (S6), Crude Petroleum Products and Natural Gas Products (S7), Non-

Table 6. The key sectors of the consumption of grey water in China in 2011–2015.

| Sector | 2011 | | 2012 | | 2013 | | 2014 | | 2015 | |
|---|---|---|---|---|---|---|---|---|---|---|
| | TI | DI | TI | DI | TI | DI | TI | DI | TI | DI |
| S1 | 0.3829 | 0.5271 | 0.3821 | 0.5222 | 0.3860 | 0.5233 | 0.3503 | 0.5061 | 0.3178 | 0.4832 |
| S18 | 0.0034 | 0.0137 | 0.0035 | 0.0137 | 0.0035 | 0.0136 | 0.0041 | 0.0137 | 0.0045 | 0.0141 |
| S83 | 0.3971 | 0.3988 | 0.4032 | 0.4047 | 0.4023 | 0.4037 | 0.4186 | 0.4197 | 0.4376 | 0.4389 |
| S88 | 0.0025 | 0.0029 | 0.0025 | 0.0029 | 0.0025 | 0.0029 | 0.0028 | 0.0030 | 0.0032 | 0.0032 |
| S89 | 0.0046 | 0.0032 | 0.0045 | 0.0032 | 0.0045 | 0.0032 | 0.0053 | 0.0033 | 0.0061 | 0.0036 |
| S90 | 0.0088 | 0.0033 | 0.0086 | 0.0032 | 0.0085 | 0.0032 | 0.0096 | 0.0033 | 0.0103 | 0.0035 |
| S92 | 0.0056 | 0.0034 | 0.0055 | 0.0033 | 0.0054 | 0.0033 | 0.0062 | 0.0033 | 0.0069 | 0.0035 |
| S95 | 0.0244 | 0.0036 | 0.0238 | 0.0035 | 0.0234 | 0.0035 | 0.0242 | 0.0036 | 0.0232 | 0.0040 |
| S99 | 0.0032 | 0.0032 | 0.0031 | 0.0032 | 0.0030 | 0.0033 | 0.0031 | 0.0033 | 0.0032 | 0.0033 |
| Total 1 | 0.8300 | 0.9563 | 0.8343 | 0.9570 | 0.8366 | 0.9571 | 0.8214 | 0.9563 | 0.8096 | 0.9541 |
| Total 2 | 0.8325 | 0.9592 | 0.8368 | 0.9599 | 0.8391 | 0.9600 | 0.8242 | 0.9593 | 0.8128 | 0.9573 |

Table 5 does not include the value of Sector 88; Table 6 includes the value of Sector 88.

**Table 7. Direct Impact (DI) dominant sectors and Total Impact (TI) dominant sectors of the blue water footprint in 2015.**

| Sectors | TI | DI | Sectors | TI | DI | Sectors | TI | DI |
|---------|------|------|---------|------|------|---------|------|------|
| s6 | 0.0123 | 0.0013 | S37 | 0.0055 | 0.0015 | s81 | 0.0019 | 0.0038 |
| s7 | 0.0187 | 0.0008 | S38 | 0.0337 | 0.0011 | S84 | 0.0279 | 0.0029 |
| S11 | 0.0046 | 0.0007 | S39 | 0.0060 | 0.0008 | S89 | 0.0075 | 0.0009 |
| S12 | 0.0035 | 0.0012 | s42 | 0.0035 | 0.0014 | s90 | 0.0152 | 0.0009 |
| s14 | 0.0029 | 0.0056 | s47 | 0.0187 | 0.0013 | S92 | 0.0104 | 0.0008 |
| s16 | 0.0018 | 0.0066 | s59 | 0.0046 | 0.0024 | s95 | 0.0395 | 0.0010 |
| s17 | 0.0016 | 0.0035 | s62 | 0.0061 | 0.0024 | s96 | 0.0512 | 0.0007 |
| S31 | 0.0037 | 0.0013 | s63 | 0.0036 | 0.0015 | s97 | 0.0080 | 0.0006 |
| S32 | 0.0033 | 0.0010 | S67 | 0.0039 | 0.0019 | s99 | 0.0031 | 0.0008 |
| S35 | 0.0078 | 0.0024 | S72 | 0.0033 | 0.0021 | Total | 0.3139 | 0.0532 |

metal Minerals and Other Mining (S11) (all of which are sectors in mining industry), and Grain Mill Products (S12), which are sectors in the food processing industry. The manufacturing industry sectors that are total impact dominant are as follows: Paper and Products (S31), Printing and Record Medium Reproduction (S32), Petroleum Refining (S35), Raw Chemical Materials (S37), Chemical Fertilizers (S38), Chemical Pesticides (S39), Chemicals for Special Usages (S42), Plastic Products (S47), Metal Products (S59), Other General Industrial Machinery (S62), Agriculture, Forestry, Animal Husbandry, and Fishing Machinery (S63), Vehicle Fittings Production (S67), Other Electric Machinery and Equipment (S72), and Electricity and Steam Production and Supply (S84). The sectors belonging to the tertiary industry that are total impact dominant sectors include Railway Freight Transport (S89), Highway Freight and Passenger Transport (S90), Water Freight and Passenger Transport (S92), Pipeline Transport (S95), Hotels (S96), Water Conservancy (S97), and Health Services (S99). S95–S97 and S99 were formed after merging; in this paper, the sectors with original non-zero blue water data were considered. It is worth noting that, as Table 1 shows, the TI of some sectors in the tertiary industry can never be large, even though they only consume a little blue water. Due to the correlation effect among the different industries, an increase in the value added of a sector in the tertiary industry by 1% will cause a large change in the use of blue water in other sectors. We combined the total values in 2015 of 36 sectors in Tables 5 and 7, and we found that the values of the TI and DI of the key sectors, total impact dominant sectors, and direct impact dominant sectors account for 91.09% and 92.26% of all 99 sectors. Thus, the remaining 63 sectors can be called non-key sectors.

Table 8 indicates that according to the standard of TI<0.003 and DI≥0.003,the direct impact dominant sectors in terms of the grey water use in China in 2015 are as follows: Domestic Public Transport (S91), Air Passenger Transport (S93), and Air Freight Transport

**Table 8. Direct Impact (DI) dominant sectors and Total Impact (TI) dominant sectors of the grey water footprint in 2015.**

| Sectors | TI | DI | Sectors | TI | DI | Sectors | TI | DI |
|---------|------|------|---------|------|------|---------|------|------|
| S4 | 0.0031 | 0.0000 | S38 | 0.0202 | 0.0003 | s93 | 0.0021 | 0.0034 |
| S5 | 0.0066 | 0.0020 | S39 | 0.0035 | 0.0002 | s94 | 0.0024 | 0.0039 |
| S6 | 0.0063 | 0.0000 | s47 | 0.0101 | 0.0001 | s96 | 0.0307 | 0.0028 |
| S7 | 0.0103 | 0.0000 | S84 | 0.0145 | 0.0001 | s97 | 0.0058 | 0.0026 |
| S35 | 0.0045 | 0.0005 | s91 | 0.0025 | 0.0031 | Total | 0.1227 | 0.0191 |

A 0.0000 DI value indicates the actual DI value of some sectors is smaller than 0.0001.

(S94), all of which belong to the tertiary industry. Based on the standard of TI≥0.003 and DI<0.003, the total impact dominant sectors of grey water use of China in 2015 include the sectors in the primary industry, namely, Livestock and Livestock Products (S4) and Fishery, Technical Services for Agriculture, Forestry, Livestock, and Fishing (S5). Sectors in the secondary industry include Coal Mining and Processing (S6), Crude Petroleum Products and Natural Gas Products (S7), Petroleum Refining (S35), Chemical Fertilizers (S38), Chemical Pesticides (S39), Plastic Products (S47), and Electricity and Steam Production and Supply (S84). Tertiary industry sectors with Hotels (S96) and Water Conservancy (S97) are included. S96 and S97 are merged sectors, and we only considered the sectors with non-zero values for their grey water use. We obtained the key sectors by combining the total values of the TI and DI of 23 sectors in 2015 in Tables 6 and 8; the total impact dominant sectors and direct impact dominant sectors account for 93.55% and 97.64% of the total impact and direct impact among all 99 sectors, so the remaining 76 sectors can be called non-key sectors.

## Cross-elasticity between water consumption and value added

We further analyzed the cross-elasticity between the water consumption and value added with Eq (13). Due to the large number of sectors involved, we took the most important key sector, Crop Cultivation, as an example to illustrate the cross-elasticity between the consumption of green water, blue water, and grey water of the Crop Cultivation sector and the value added of key sectors such as Crop Cultivation (S1), Forestry (S2), Livestock and Livestock Products (S4), Fishery, Technical Services for Agriculture, Forestry, Livestock, and Fishing (S5), Other Food Products (S18), and Scrap and Waste (S83). The specific results are shown in Table 9.

The row values in Table 9 correspond to the TI of the Crop Cultivation sector in Table 3, and the column values correspond to the DI of the sector Crop Cultivation sector in Table 3. According to the corresponding correlativity, the total row value of the cross-elasticity between the total water consumption in the Sector Crop Cultivation sector and the value added of the key sectors in 2011–2015 can explain 98.86% (0.5333/0.5394), 98.90%(0.5359/0.5419), 98.92% (0.5412/0.5471), 99.01% (0.5003/0.5053), and 99.08% (0.4638/0.4681) of the TI, while correspondingly, the total column value can explain 73.23% (0.5343 /0.7296), 73.74% (0.5367/ 0.7278), 74.31% (0.5417/0.7290), 70.12% (0.5036/0.7182), and 66.96% (0.4693/0.7008) of the DI, which indicates the cross-elasticity between the total consumption of water resources by Crop Cultivation sector and the value added of other non-key sectors is very small.

In addition, the row and column value of 0.5236 for the Crop Cultivation (S1) sector means that a 1% increase in the value added of the Crop Cultivation (S1) sector in 2011 causes an increase in its water consumption by 0.5236%. The row value of Livestock and Livestock Products (S4) is 0.0033, which represents a 1% increase in the value added by Livestock and Livestock Products (S4) that results in a 0.0033% increase in the water use of Crop Cultivation (S1). The column value of the Livestock and Livestock Products (S4) sector is 0.0030, which indicates a 1% increase in the value added of Crop Cultivation (S1) that leads to a 0.0030% increase in the water consumption for Livestock and Livestock Products (S4).

To analyze the cross-elasticity between the water consumption and the value added of various sectors, we took the total consumption of the green water, blue water, and grey water in 2015 as an example (similar analysis for other years)to conduct the network analysis based on the 99 × 99 matrix constructed by cross-elasticity between water resource consumption of each type and the value added. Specific results are shown in Fig 2.

Fig 2 is a 1-mode network formed by the data of the normalization of representative elements $W_{ij}^v$ of a matrix $\mathbf{W^v}$. The process of data normalization is as follows: if $W_{ij}^v \geq 1/(99 \times 99)$, then it is 1, otherwise it is 0. S1→S2 is an example illustrating the meaning

**Table 9. The cross-elasticity of the consumption of green water, blue water, and grey water of Crop Cultivation (sector 1) with the value added of key sectors.**

| Sectors | 2011 | | 2012 | | 2013 | | 2014 | | 2015 | |
|---|---|---|---|---|---|---|---|---|---|---|
| | Row | Column | Row | Column | Row | Column | Row | Column | Row | Column |
| S1 | 0.5236 | 0.5236 | 0.5263 | 0.5263 | 0.5316 | 0.5316 | 0.4920 | 0.4920 | 0.4566 | 0.4566 |
| S2 | 0.0000 | 0.0009 | 0.0000 | 0.0009 | 0.0000 | 0.0008 | 0.0000 | 0.0009 | 0.0000 | 0.0009 |
| S4 | 0.0033 | 0.0030 | 0.0033 | 0.0029 | 0.0033 | 0.0029 | 0.0030 | 0.0030 | 0.0027 | 0.0030 |
| S5 | 0.0003 | 0.0049 | 0.0003 | 0.0048 | 0.0003 | 0.0047 | 0.0003 | 0.0057 | 0.0002 | 0.0068 |
| S18 | 0.0059 | 0.0004 | 0.0059 | 0.0004 | 0.0058 | 0.0004 | 0.0049 | 0.0004 | 0.0042 | 0.0005 |
| S83 | 0.0001 | 0.0015 | 0.0001 | 0.0014 | 0.0001 | 0.0014 | 0.0001 | 0.0015 | 0.0001 | 0.0015 |
| Total | 0.5333 | 0.5343 | 0.5359 | 0.5367 | 0.5412 | 0.5417 | 0.5003 | 0.5036 | 0.4638 | 0.4693 |

A 0.0000 cross-elasticity value indicates that the cross-elasticity of consumption of the green water, blue water, and grey water of Crop Cultivation with the value added of the corresponding key sectors is smaller than 0.0001.

of the arrow direction in Fig 2: S1→S2 represents the 1% increase in the value added by Forestry (S2) that causes the percentage increase in the water consumption of Crop Cultivation (S1) to be larger than $1/(99 \times 99)$%. Fig 2 shows that there are more connections between the key sectors of water consumption, such as Crop Cultivation (S1) and other sectors, which indicates that the cross-elasticity of the value added with the water consumption between the key sectors and other sectors is greater. In addition, there is no connection between Other Textiles Not Elsewhere Classified (S25), Toys, Sporting, Athletic, and Recreational Products (S34),

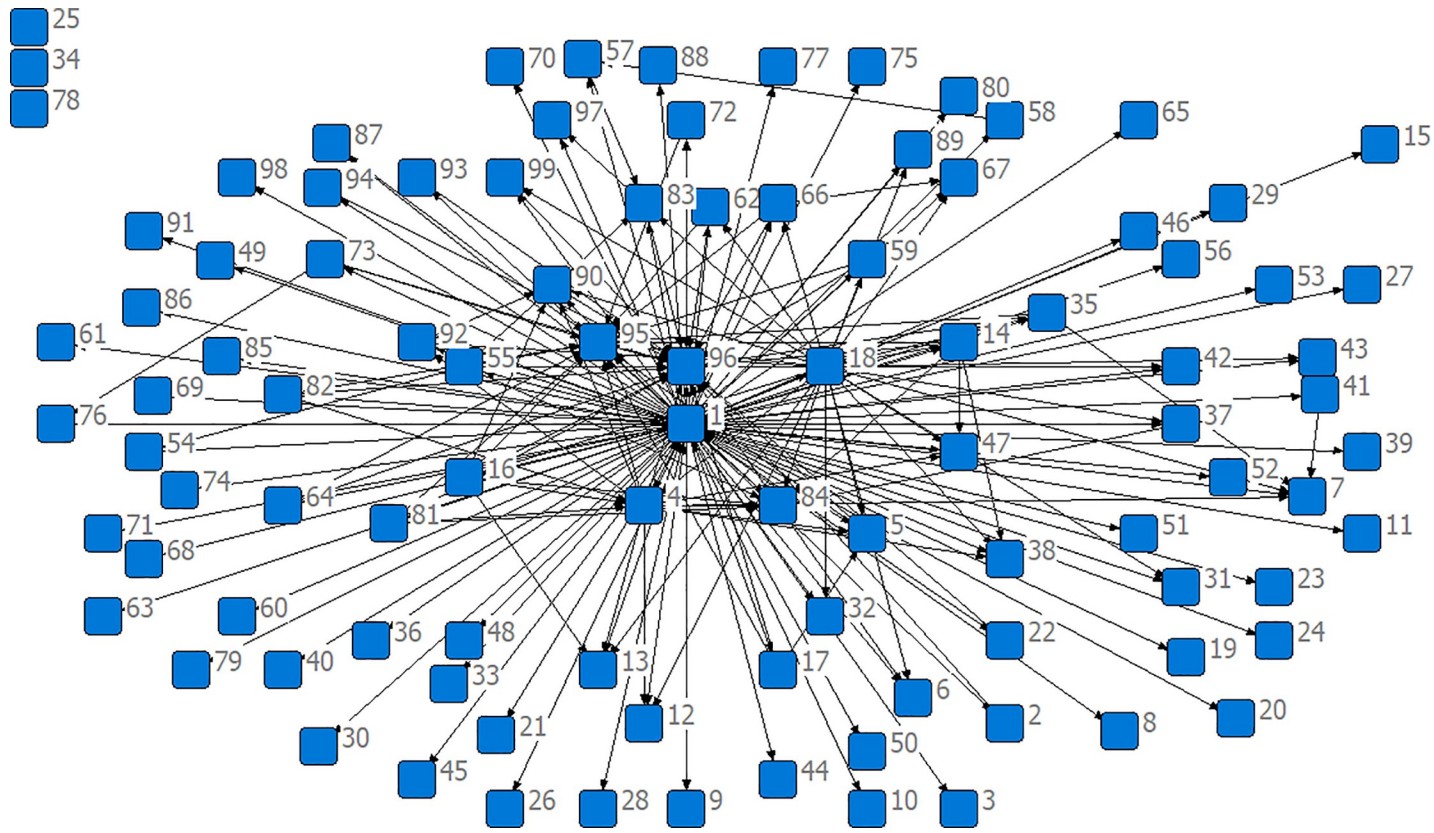

**Fig 2. The cross-elasticity of the total consumption of the green water, blue water, and grey water with the value added of various sectors in 2015.**

Other Electronic and Communication Equipment (S78), and other sectors because the cross-elasticity of the water consumption with the value added between these sectors and others is less than the threshold (1/(99 x 99)).

## Further discussion

In recent years, with the rapid development of the Chinese economy, the imbalance between supply and demand of resources in China is becoming more serious, and the environmental pressure is also growing. A significant amount of research has been done to explore the problem of environmental pollution in China from the perspective of the carbon footprint [37–42]. This study explores the disparity between the supply and demand of water resources in China from the perspective of the water footprint.

Since the raw data collected from different researchers is not quite the same, there are different accounting results of China's water footprint, even with the same calculation method. For example, there are differences in terms of water footprint accounting results between this paper and the existing literature on studies of the water footprint for each sector in China[2], [3] [19–21] [23], and there are differences among the studies within the literature. In the literature, water footprint accounting in each region of China is based on the multi-region input–output model [1] [27–29] [34]; the sum of the water footprint in each region is not equal to China's water footprint in this paper. Additionally, the focus of most studies of China's water footprint with the input–output model encompasses the green water footprint and blue water footprint, while only a few [19] are about China's green water, blue water, and grey water footprint altogether.

Several studies [2] [23] stressed the impact of the virtual water trade and water footprint, but the focus of this paper is the analysis of the key sectors of the water resource consumption in China. Based on the cross-elasticity of the value added with the water consumption, we defined the DI indicator and TI indicator as methods to identify key sectors of the water consumption instead of considering the amount of water used for each sector alone. Therefore, not only do the key sectors identified in this paper require a large amount of water, but their economic growth also has a relatively significant impact on the water resource use of other sectors. Table 4 shows that the direct impact dominant sectors of China's water consumption in 2015 are Vegetable Oil and Forage (S14), Slaughtering, Meat Processing, Eggs, and Dairy Products (S16), Prepared Fish and Seafood (S17), and Arts and Crafts Products(S81). Though these sectors used a relatively large amount of water (DI>0.003), the impact of their economic growth on the water consumption of other sectors was comparatively small (TI<0.003). Therefore, they were not ranked as key sectors of China's water consumption in 2015.

In terms of the sum of the green water, blue water, and grey water footprint, key sectors of China's water consumption in 2015 are Crop cultivation(S1), Forestry(S2), Livestock and livestock products(S4), Fishery, Technical Services for Agriculture, Forestry, Livestock and Fishing(S5), Other Food Products(S8), and Scrap and Waste(S83). These sectors belong to the Agriculture, Food Processing Industry, and Wastewater Treatment Industry. Farming consumes a large amount of water, and farm products are necessary raw materials for production in other industries. Therefore, the Direct Impact (DI) indicator and Total Impact (TI) indicator are relatively great for Crop Cultivation (S1), Forestry(S2), Livestock and Livestock Products(S4), Fishery, and Technical Services for Agriculture, Forestry, Livestock, and Fishing(S5). As the food processing industry is closely related to farm products, the Direct Impact (DI) and Total Impact (TI) for Other Food Products (S18) are relatively great as well. Wastewater treatment industry mainly involves the grey water footprint. To purify wastewater, a significant amount of water is required. As the production of most sectors results in wastewater,

wastewater treatment cannot be avoided. The full impact of an increase in the value added for Scrap and Waste (S83) on water consumption of other industries is relatively large. These explanations apply to the results from Table 3 and Table 9, so they will not be mentioned again.

Key sectors of China's water use vary from year to year. For instance, according to Table 2, for sector S2 (Forestry), the DI indicator in 2011–2013 is less than 0.003, which does not conform to TI>0.003 and DI>0.003, so it cannot be identified as a key sector. However, in 2014–2015, sector S2 (Forestry) meets TI>0.003 and DI>0.003, and thus, it is identified as a key sector. In the EORA database, the water usage of each sector in China varies from year to year, and there are different input–output tables each year. Therefore, even for the same sector, the DI indicator and TI indicator vary from year to year. Accordingly, the key sectors of China's water use might change in different years.

Key sectors of China's water consumption might also change when it comes to different types of water footprints. For instance, according to Table 5 and Table 6, the key sectors that use blue water are different from those with grey water. As the functions of blue water and grey water are not the same, one for the production and services of each sector while the other for removing pollutants from water, sectors that use more blue water do not necessarily consume more grey water. In addition, how blue water use of other sectors changes, caused by the change in the value added of each sector, is different from the change in grey water use.

## Conclusions and implications

In order to make a more rational use of water resources, we identified the key sectors of water resources utilization in China from the perspective of the water footprint. The empirical results from a comprehensive analysis of the use of green water, blue water, and grey water show that the key sectors for water consumption in China in 2015 were Crop Cultivation (S1), Forestry (S2), Livestock and Livestock Products (S4), Fishery, Technical Services for Agriculture, Forestry, Livestock, and Fishing (S5), Other Food Products (S18), and Scrap and Waste (S83). The above six sectors accounted for 66.15% of the total impact and also explained the direct impact by 90.76%. Steel Processing (S55) is another key sector due to its blue water use. In 2015, the seven key sectors of blue water consumption in China explained 59.70% of the total impact and 86.94% of the direct effect. The key sectors for the use of grey water were Crop Cultivation (S1), Other Food Products (S18), Scrap and Waste (S83), Railway Freight Transport (S89), Highway Freight and Passenger Transport (S90), Water Freight and Passengers Transport (S92), Pipeline Transport (S95), and Health Services (S99), while the key sectors of grey water consumption of China in 2015 explained 81.28% of the total impact and 95.73% of the direct impact. In addition to the key sectors, there were a large number of direct-effect-based sectors, total-effects-based sectors, and non-key sectors.

Based on our findings, we obtained the following policy implications: production in key sectors not only drives the increase of the water consumption in other sectors but also significantly increases the water demand in key sectors and the value added of other sectors. Therefore, the Chinese government should mainly focus on the key sectors when designing water-saving policies and improving water-use efficiency, such as promoting water-saving irrigation technology, including sprinkler irrigation and drip irrigation in the agricultural sector. In addition, there were differences between the blue water and grey water consumption in the key sectors. For the key sectors of the blue water consumption, the focus should be the implementation of water-saving measures, while for the key sectors of grey water use, it is necessary to strictly implement the emission reduction measures and increase the intensity of sewage treatment. For the direct impact dominant sectors, the increase in the value added of the

economic system will have a great impact on the water consumption of these sectors. There-fore, if the water consumption of these sectors is restricted, it is not only of no service to the increase in the value added of the entire economic system, but also not conducive to the eco-nomic growth. For the total impact dominant sector, the increase in the value added of these sectors will lead to a great increase in the use of water resources in the entire economic system. From the perspective of water conservation, the economic growth of these sectors can be mod-erately controlled. Since non-key sectors are less relevant to water consumption, the authori-ties may pay less attention to them.

Sustainable water resource use is a challenging task faced by all countries, and this is partic-ularly significant in China, as China is the subject to rapid economic development and contin-uous population growth. In order to alleviate the growing imbalance between the water supply and demand in China, we offer some strategies and measures that may help address the chal-lenges, which include the following: (1) For industrial and agricultural water use, a system of applying for water, quota water supply, paid water supply, over-quota controlling, or double charging can be implemented. (2) In areas where water resources are scarce, low-water-con-suming industries should be developed and water-recycling systems should be adopted. (3) Strategies, such as developing water-saving agriculture, preventing flood irrigation, reducing canal leakage, and popularizing water-saving technologies (such as sprinkler irrigation, drip irrigation, and infiltration irrigation), should be taken.

## Supporting information

**S1 Appendix. Sector list.**
(DOC)

## Acknowledgments

We thank LetPub (www.letpub.com) for its linguistic assistance during the preparation of this manuscript.

## Author Contributions

**Data curation:** Fengying Lu.

**Writing – original draft:** Guangyao Deng.

**Writing – review & editing:** Xiaofang Yue, Lu Miao.

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
