## [Decision Letter · Decision Letter 0]

25 Nov 2019

PONE-D-19-27097

Identification of key sectors of water resource utilization in China from the perspective of the water footprint

PLOS ONE

Dear Dr Yue,

Thank you for submitting your manuscript to PLOS ONE. After careful consideration, we feel that it has merit but does not fully meet PLOS ONE’s publication criteria as it currently stands. Therefore, we invite you to submit a revised version of the manuscript that addresses the points raised during the review process.

We would appreciate receiving your revised manuscript by Jan 09 2020 11:59PM. To enhance the reproducibility of your results, we recommend that if applicable you deposit your laboratory protocols in protocols.io, where a protocol can be assigned its own identifier (DOI) such that it can be cited independently in the future. For instructions see: http://journals.plos.org/plosone/s/submission-guidelines#loc-laboratory-protocols

We look forward to receiving your revised manuscript.

Kind regards,

Chuanwang Sun

Academic Editor

PLOS ONE

Journal Requirements:

'This work was supported by the Natural Science Foundation of China under

Grant [number 71704070]; Ministry of Education for the Humanities and Social Sciences

Research Young Fund on the West and Borderland Project [number 17XJC790002], Gansu

Provincial Higher Education Research Project [number 2017B-41]; Guangdong Provincial Natural

Science Foundation of China under Grant [number 2017A030313443];Program of Lanzhou

University ofFinance and Economics under Grant [number Lzufe2018B-06]; and the Research

Project of Institute of Silk RoadEconomics, Lanzhou University of Finance and Economics under

Grant[number JYYZ201603].'

'Yes'

4. Please upload a copy of Figure 3, to which you refer in your text on page 21. If the figure is no longer to be included as part of the submission please remove all reference to it within the text.

Additional Editor Comments (if provided):

Reviewers' comments:

Reviewer #1: The paper studied the China’s water by drawing on the model from Alcantara and Padilla[7]. Furthermore, the authors spent much time collecting data and writing the paper. However, the authors did not show that why the subject was selected and what problem and question of China’s water the paper would be addressed. There are some problems which would be corrected by the authors as follows:

1.Writing skill in English should be improved,Chinese characters can even be found in this paper.

2.Pay attention to the format, for example,”Chinafrom”,”footprintand”.(Page3,Line 14)

3.I suggest Figure 2 be more clear and beautiful to facilitate the readers.

4.The logic of literature review is a little confused. Are there no relevant researches on identification of key sectors of water resource utilization in China? What is the innovation of this paper?

5.The author employs 0.003 as the threshold to classify all sectors in this paper. However, the valid reasons why is 0.003 suitable fail to be provided.

6.The author mainly describes the measurement results in this paper. But further discussions on this results find no appearance.

7.The author mentions that Chinese government should design water-saving policies and control the economic growth of key sectors.However,concrete methods also can not be found.

Reviewer #2: It is interesting to analyze water resource utilization from the perspective of water footprint. It is better for the authors to conduct MAJOR REVISION.The comments are as follows.

1. what is water footprint? why did the authors analyze from the perspective of water footprint? It is better to clarify in detail.

2. What is the motivation of this paper?

3. Compared with previous studies, what are the novelties of this paper?

4. In literature reviews, it is better for the author to compare the previous papers on water resource and other resource such coal, gas and others such as:

[1] Decoupling CO2 emissions from economic growth in agricultural sector across 30 Chinese provinces from 1997 to 2014. Journal of Cleaner Production, Volume 159, 15 August 2017, Pages 220-228

[2] Nonrenewable energy, renewable energy, carbon dioxide emissions and economic growth in China from 1952 to 2012. Renewable and sustainable energy reviews, ,2015, 52：680-688.

[3] The Comparison Analysis of Total Factor Productivity and Eco-efficiency in China’s Cement Manufactures. Energy Policy. 81:61-66,,2015

[4] Are Stronger Environmental Regulations Effective in Practice? The Case of China’s Accession to the WTO , Journal of Cleaner Production , (39):161-167，2013.

[10] The influencing factors of CO2 emission intensity of Chinese agriculture from 1997 to 2014, Environmental Science and Pollution Research,2018， 25:13093–13101

[5] Different impacts of export and import on carbon emissions across 7 ASEAN countries: A panel quantile regression approach，Science of the Total Environment，2019，686, 1019-1029

5.what are the differences among green water, blue water and grey water. It is better to introduce in detail.

6.The paper also compare different industry sectors. what are the differences?

7. In conclusion, please compare with previous studies. what are the similarities and differences?

The English language need to revise carefully.

---

## [Author Response · Author response to Decision Letter 0]

6 Jan 2020

Dear reviewers:

Thank you for your comments on our manuscript entitled “Identification of key sectors of water resource utilization in China from the perspective of the water footprint” (PONE-D-19-27097). Those comments are very helpful for revising and improving our paper, as well as the important guiding significance to other research. We have studied the comments carefully and made corrections which we hope meet with approval.See the attached attachment for specific responses

---

## [Decision Letter · Decision Letter 1]

10 Feb 2020

PONE-D-19-27097R1

Identification of key sectors of water resource utilization in China from the perspective of the water footprint

PLOS ONE

Dear Dr Yue,

Thank you for submitting your manuscript to PLOS ONE. After careful consideration, we feel that it has merit but does not fully meet PLOS ONE’s publication criteria as it currently stands. Therefore, we invite you to submit a revised version of the manuscript that addresses the points raised during the review process.

We would appreciate receiving your revised manuscript by Mar 26 2020 11:59PM. To enhance the reproducibility of your results, we recommend that if applicable you deposit your laboratory protocols in protocols.io, where a protocol can be assigned its own identifier (DOI) such that it can be cited independently in the future. For instructions see: http://journals.plos.org/plosone/s/submission-guidelines#loc-laboratory-protocols

We look forward to receiving your revised manuscript.

Kind regards,

Chuanwang Sun

Academic Editor

PLOS ONE

Editor: Please polish the language before the next submission.

Reviewer #1: It is reasonable to study key sectors of water resource utilization in China. The authors completely answered the questions from the reviewer.There two questions required to response.

1.The English of the paper need to be revised again.

2.the paper should cite more references. For example, Spatio-temporal analysis of driving factors of water resources consumption in China

Reviewer #2: Compared with the previous version, the quality of this paper has improved greatly.

I recommend to accept this paper.

---

## [Decision Letter · Decision Letter 2]

22 Apr 2020

PONE-D-19-27097R2

Identification of key sectors of water resource utilization in China from the perspective of the water footprint

PLOS ONE

Dear Dr Yue,

Thank you for submitting your manuscript to PLOS ONE. After careful consideration, we feel that it has merit but does not fully meet PLOS ONE’s publication criteria as it currently stands. Therefore, we invite you to submit a revised version of the manuscript that addresses the points raised during the review process. Especially, there are still a few writing errors, I suggest that the language should be polished by the language editing service.

We would appreciate receiving your revised manuscript by Jun 06 2020 11:59PM. To enhance the reproducibility of your results, we recommend that if applicable you deposit your laboratory protocols in protocols.io, where a protocol can be assigned its own identifier (DOI) such that it can be cited independently in the future. For instructions see: http://journals.plos.org/plosone/s/submission-guidelines#loc-laboratory-protocols

We look forward to receiving your revised manuscript.

Kind regards,

Chuanwang Sun

Academic Editor

PLOS ONE

Reviewers' comments:

Reviewer's Responses to Questions

Reviewer #2: compared with previous studies, the quality of this paper has improved greatly.

I recommend to accept this paper.

Reviewer #3: The article has been revised perfectly, the content is innovative, the angle is scientific and the method is feasible. Only one point: some fresh paper can be added as reference, eg:

Lu, S., Wu, X., Sun, H. et al. The multi-user evolutionary game simulation in water quality-based water source system. Environ Geochem Health (2019). https://doi.org/10.1007/s10653-019-00315-5.

---

## [Decision Letter · Decision Letter 3]

26 May 2020

Identification of key sectors of water resource utilization in China from the perspective of the water footprint

PONE-D-19-27097R3

Dear Dr. Yue,

We are pleased to inform you that your manuscript has been judged scientifically suitable for publication and will be formally accepted for publication once it complies with all outstanding technical requirements.

With kind regards,

Chuanwang Sun

Academic Editor

PLOS ONE

---

## [Editor Report · Acceptance letter]

11 Jun 2020

PONE-D-19-27097R3 

Identification of key sectors of water resource utilization in China from the perspective of the water footprint 

Dear Dr. Yue:

I'm pleased to inform you that your manuscript has been deemed suitable for publication in PLOS ONE. Congratulations! Your manuscript is now with our production department. 

Kind regards, 

on behalf of

Dr. Chuanwang Sun 

Academic Editor

PLOS ONE